# Osteosynthesis Metal Plate System for Bone Fixation Using Bicortical Screws: Numerical–Experimental Characterization

**DOI:** 10.3390/biology11060940

**Published:** 2022-06-20

**Authors:** Andrea A. R. Olmos, Aureliano Fertuzinhos, Teresa D. Campos, Isabel R. Dias, Carlos A. Viegas, Fábio A. M. Pereira, Nguyễn T. Quyền, Marcelo F. S. F. de Moura, Andrea Zille, Nuno Dourado

**Affiliations:** 1CMEMS-UMinho, Universidade do Minho, 4800-058 Guimarães, Portugal; andrea.olmos@dem.uminho.pt (A.A.R.O.); afertuzinhos@dem.uminho.pt (A.F.); teresa.ac.biome@gmail.com (T.D.C.); 2LABBELS—Associate Laboratory, Braga, 4800-122 Guimarães, Portugal; 3CECAV—Centro de Ciência Animal e Veterinária, Universidade de Trás-os-Montes e Alto Douro, 5001-801 Vila Real, Portugal; idias@utad.pt (I.R.D.); cviegas@utad.pt (C.A.V.); 4AL4AnimalS—Laboratório Associado para Ciência Animal e Veterinária, 1300-477 Lisboa, Portugal; 5CITAB, Universidade de Trás-os-Montes e Alto Douro, 5001-801 Vila Real, Portugal; famp@utad.pt; 62C2T—Centro de Ciência e Tecnologia Têxtil, Universidade do Minho, 4800-058 Guimarães, Portugal; quyen@2c2t.uminho.pt (N.T.Q.); azille@2c2t.uminho.pt (A.Z.); 7Faculdade de Engenharia, Departamento de Engenharia Mecânica, Universidade do Porto, 4200-465 Porto, Portugal; mfmoura@fe.up.pt

**Keywords:** cortical bone tissue, bone fracture, cohesive zone modeling, finite element model, bone–screw interface

## Abstract

**Simple Summary:**

This study addresses an important issue concerning the evaluation of stresses in bone shafts stabilized by osteosynthesis metal plates, following routine surgical procedures to repair severe fractures in bone. It is recognized that bone regeneration following fracture is highly dictated by the stress state in the damaged regions. Since metallic inserts, like plates and screws, are usually employed to assure the stabilization of fractures in bone, it is important to evaluate the effect of those parts on the developed stresses in bone tissue. In the present work fracture was induced in a femoral bone of an animal model, which was suitably stabilized with a dynamic compression plate (DCP) using bicortical screws. This system was submitted to bending to trigger damage in bone tissue in the vicinity of metal inserts. Finite element modelling was then performed to mimic damage initiation and propagation in bone, thus simulating the results observed experimentally. Stress distributions in the vicinity of the screwed regions due to fastening of DCP allowed to identify very significant differences, which can affect bone hilling processes. It can be concluded that the developed procedure may be used to help surgeons to support decisions regarding bone repair using standard DCP.

**Abstract:**

This study reports the numerical and experimental characterization of a standard immobilization system currently being used to treat simple oblique bone fractures of femoral diaphyses. The procedure focuses on the assessment of the mechanical behavior of a bone stabilized with a dynamic compression plate (DCP) in a neutralization function, associated to a lag screw, fastened with surgical screws. The non-linear behavior of cortical bone tissue was revealed through four-point bending tests, from which damage initiation and propagation occurred. Since screw loosening was visible during the loading process, damage parameters were measured experimentally in independent pull-out tests. A realistic numerical model of the DCP-femur setup was constructed, combining the evaluated damage parameters and contact pairs. A mixed-mode (I+II) trapezoidal damage law was employed to mimic the mechanical behavior of both the screw–bone interface and bone fractures. The numerical model replicated the global behavior observed experimentally, which was visible by the initial stiffness and the ability to preview the first loading peak, and bone crack satisfactorily.

## 1. Introduction

Long bones, such as the humerus, femur, and tibias, have been widely studied in the past few years to evaluate the mechanical response of classical bone models in different loading environments. This allowed the development of new implant materials mechanically comparable to human bones regarding the ability to dissipate energy and withstand different physiological loads. In the majority of those studies, cortical bone tissue was considered as the reference material due to its superior mechanical properties. Cortical bone tissue is an anisotropic, complex hierarchical tissue composed of high mineral content that contributes to the characteristic stiffness and hardness [1]. The diaphysis of a long bone is mainly composed of cortical bone. The outer surface is covered by periosteum, which has the potential to form bone during growth and fracture healing [1]. Fractures can result from daily activities and trauma when the bone’s mechanical strength is exceeded due to applied external forces [2]. Bone repair is performed using various mechanical fixation systems that employ wires, screws, pins, and/or metal plates. The literature reports clinical problems in conventional internal fixation techniques, wherefore, functional or geometrically improvements are required to reduce malfunctions [3]. Osteosynthesis metal plates associated to bicortical screws, contoured to the compressive face of long bones, are examples frequently used to promote fracture healing, offering the required mechanical stabilization and fracture alignment. Since the disposition of bicortical screws can induce important stress concentration in the screw–bone interface, the evaluation of damage parameters in this region is fundamental. In fact, MacLeod et al. [4] considered that this interface has a considerable impact on the local stress–strain fields in the vicinity of the screws, leading to different mechanical responses of bone damage and ultimately screw loosening. The combination of experimentally obtained data with numerical analyses allowed measuring local bone damage parameters with accuracy. Stress and strain fields in bone tissue result from the application of experimentally equivalent external loads [5,6], providing that constitutive material laws have been correctly adopted. Avery et al. [5] demonstrated the strengthening effect of a prophylactic internal fixation system, using sheep tibia models, commonly used as a human model, and four different configurations of plates for stabilization. Some advances in the design of immobilization systems have included lighter and thinner plates with limited contact areas that reduce the risk of osteopenia and stress shielding [5], different screw configurations [7,8], or screw positioning [9], to stabilize either transversal or comminuted fractures.

Several studies report that stiffness and fracture failure highly depend on the screw location near the fracture region [10], as well as the screw type or screw positioning [11,12]. By ranging the fixation system with the bone shaft, it is possible to modify the interfragmentary movement at the fracture gaps, thus improving the quality of the fracture healing. From a clinical point of view, a plate with a small number of locking screws may reduce damage both on bone and soft tissues, requiring minimal surgical access for fracture stabilization. On the other hand, a large number of locking screws can enhance the fixation stability but ultimately lead to stress-shielding effects [9]. These observations demonstrate the clinical relevance to develop adequate tools to analyze the effects of cortical bone fastening combined with metal plates [9,13]. However, few studies have focused on the analysis of bone fracture properties to understand the influence of bone damage initiation and propagation in the fracture pattern, as well as on the development of new biomimetic materials. 

MacLeod et al. [4] found that the interface modeling strategy has almost no influence on the global load–displacement behavior, contradicting the study of Karunratanakul et al. [14], who found stiffness variations among different fixation systems. However, these studies have few (or no) experimental validations, are limited by the adopted simplification strategies (e.g., symmetry conditions) for the screw positioning and bone geometry, and do not account for the existence of fractures in bones due to stress concentration [14]. Faced with so many studies dedicated to bone screws, as well as different plate working lengths and bone–plate–screw interfaces, other concerns are justified regarding the analysis of the influence of structural parameters on bone fixation systems, e.g., metal plates [2,15]. In this sense, Sheng et al. [2] performed a finite element analysis (FEA) with different structural parameters of a locking plate based on the combination of uniform and orthogonal design and taking into account the fixation stability and healing performance as the criteria. 

In this study, the mechanical performance of a standard dynamic compression plate (DCP) model, used to stabilize a simple oblique fracture in the neutralization function and associated to a lag screw, was performed through cohesive zone modeling (CZM). Pull-out strength (POS) of cortical screws was evaluated since screw loosening is clearly visible in immobilization systems formed by metal plates when submitted to loading. Four-point bending (FPB) tests were executed to evaluate the bending strength of the repaired structure. The aim was to present a numerical methodology that can be used in the design of bone repair systems employing standard plates and, posteriorly, locking plates. Realistic configurations of bone repairing systems were mimicked through a numerical approach based on the mixed-mode I+II+III cohesive zone model. The developed model was able to mimic both the screw-loosening phenomenon and damage (cracks) in the vicinity of screws.

## 2. Materials and Methods

Biological samples were prepared with the required geometrical configurations to perform several mechanical tests, namely three-point bending (TPB), POS, and FPB tests, described in detail in the next section. The TPB was performed to evaluate the elastic modulus of goat bone cortical tissue in the longitudinal direction. The POS test allowed to evaluate the parameters of a cohesive law that mimics the mechanical behavior of the screw–bone interface under loading. The FPB test was chosen as a test model to characterize the bending behavior of a selected immobilization system to repair a common fracture of the femoral shaft. To this aim, femurs were harvested from healthy Serrana breed goats (*Capra aegagrus hircus*) (4 years old), provided by ANCRAS (National Association of the Serrana Goat Breed), following euthanasia. Cortical bone samples were wrapped in gauzes duly moistened with saline solution and then frozen at −20 °C until testing. These procedures were performed in an operating theatre sited in the Veterinary Teaching Hospital at the University of Trás-os-Montes e Alto Douro (UTAD), following both hygienic and safety standards.

### 2.1. Three-Point Bending Specimens

The epiphysis and metaphysis of goat femurs were separated from the diaphysis using a pneumatic surgical motor system. A longitudinal cut was performed in the diaphysis of the femur, resulting in its separation into two bone fragments (Figure 1a). Each bone section was flattened using a polishing machine from Struers^®^, Cleveland, OH, USA (model ROTOPOL), resulting in regular specimens (Figure 1b). 

### 2.2. Pull-Out Specimens

Epiphyses were first removed from the femoral diaphysis cortex. Three osteotomies were executed in the diaphysis region to obtain four identical bone samples (30 mm in length each). Each sample was then radially drilled in the central section and thread-milled to allow fastening cortical screws (nominal diameter of 4.5 mm; Synthes, AO^®^, Solothurn, Switzerland) through the cortical wall and bone marrow (Figure 2).

### 2.3. Four-Point Bending Specimens

Osteotomies were performed in femur shafts using a surgical motor system to mimic an oblique fracture in the central region (Figure 3). The fixation of metal plates (standard 4.5 mm broad DCP; Synthes, AO^®^, Solothurn, Switzerland) was preceded by femur thawing at room temperature (21–23 °C) while keeping the bone hydrated in saline solution. Osteosynthesis plates (OP) were previously contoured to the lateral face of the femur, rendering possible the correct alignment of bone segments. This procedure followed surgical protocols currently executed to allow adequate bone healing. Six threaded holes were executed along the femur shaft, and one perpendicularly to the central oblique fracture to fasten cortical screws. A 4.5 mm broad DCP plate with 8-holes was applied using six bicortical screws (4.5 mm of nominal diameter; Synthes, AO^®^), in a neutralization function, subsequent to thread-milling operations. A lag screw (3.5 mm of nominal diameter; Synthes, AO^®^), placed perpendicularly to the fracture line and externally to the DCP, was fixed in such a way as to ensure mechanical stabilization of the oblique fracture, while assuring the dynamic compression along the fracture line (Figure 4).

## 3. Mechanical Tests

The numerical characterization of a fractured femoral shaft stabilized by a standard DCP and bicortical screws was envisaged in this study. With this goal, three mechanical tests were previously executed: (a) TPB, (b) POS and (c) FPB tests. The TPB test was executed to evaluate the elastic modulus of goat cortical bone tissue along the longitudinal direction. The POS test was performed to accurately identify damage parameters necessary to characterize the connection at the bone–screw interface. The FPB test was executed to evaluate the bending strength of the repaired structure.

### 3.1. Three-Point Bending Tests

Three-point bending (TPB) tests were performed to determine the longitudinal elastic modulus ($E_{L}$) of goat cortical bone beams, using the theory of Bernoulli–Euler for beams,
(1)$$E_{L}=\frac{R\;L^{3}}{4\;B\;h^{3}}$$
where R stands for the bending stiffness, L for the loading span, B for the width of the specimen cross-section, and h for the height.

A servo-electrical testing system (MicroTester INSTRON^®^ 5848) was used considering a constant displacement rate of 0.5 mm/min. Two specimens with similar dimensions (i. e.,  L=55 ,  h=7.5 ,  B=2 ;  L=55 ,  h=6.6 ,  B=1.9) were tested to obtain load–displacement curves.

### 3.2. Pull-Out Strength Tests

Pull-out strength (POS) tests were performed in four shaft sections harvested from goat femurs in coincident positions to the ones used to fix the osteosynthesis plate. During the test, bone samples were immobilized by a metal plate rigidly fixed by two screws fastened to a base (Figure 5a). The bicortical screw was pulled out by its head, which was fixed to a steel U-profile and a dowel. A universal testing machine (INSTRON^®^ 4208), equipped with a 5 kN load cell was used to conduct the pull-out tests at 0.5 mm/min. The adopted experimental protocol ensured that the energy transmitted to the mechanical system was mostly dissipated in the form of damage in the bone threaded region (Figure 5b). The experimental data (P−δ curve) from this test allowed the identification of cohesive parameters required to characterize the connection at the bone–screw interface. 

### 3.3. Four-Point Bending Tests

Four-point bending (FPB) tests were carried out in a servo-hydraulic testing system (INSTRON^®^ 8801) (Figure 6), under displacement control (0.5 mm/min). Four steel cylinders were used as supports and loading devices, considering a loading span of 110 mm (bottom supports) and 39 mm between the top-loading cylinders. Six specimens were tested in these conditions to obtain the load–displacement curves.

## 4. Numerical Models

A numerical analysis of the POS and FPB tests, including a cohesive zone model (CZM), was implemented aiming to simulate the initiation and propagation of the damage in cortical bone tissue. Thus, successive parallel fine cuts were executed along the longitudinal axis of the bone shaft, subsequent to the mechanical tests. Next, digital images of bone cross-sections were captured, keeping the original orientation of the bone. Then, solid models (CAD) were constructed from the referred images using homemade numerical tools developed in MatLab^®^, from a sequence of contours of the referred cross-sections. The obtained solid models of bone samples were then used to construct FE meshes for POS and FPB tests (Figure 7 and Figure 8). The models for the POS test simulation were constructed from bone samples extracted from four different regions along the axis of the goat femoral shaft, as represented in Figure 8a (approximately 30 mm in length each). Those regions were selected to evaluate the cohesive law of the interface cortical bone screw along the femoral diaphysis according to the experimental protocol used for the pull-out test. These regions of cortical bone (CB) tissue were modeled with, on average, 5160 8-node linear hexahedral isoparametric elements. The remaining parts, namely the screw shaft, screw head, and metal plate were modeled with 208 8-node linear hexahedral isoparametric, 96 6-node linear triangular prisms, and 1320 4-node tetrahedral elements, respectively. Two-dimensional 16 8-node cohesive elements (CE) were disposed along the screw shaft in contact with the cortical bone to mimic the screw–bone interface. Contact pairs were defined with rigid interaction between the screw and metal plate to prevent interpenetration in the course of the loading process while softening contact was modeled between the screw and the bone sample, as well as with the metal plate and bone sample.

The bone shaft for the FPB test (Figure 8b) was modeled in two halves with 133,536 6-node linear triangular prism elements, while the DCP plate and the four cylinders were modeled with 77,222 4-node tetrahedral elements and 576 8-node brick elements, respectively. The bicortical screws (six to fasten the plate and one for the lag) were modeled in the same way as for the POS test. As for the interfaces of screw-cortical bone, a total of 765 6-node cohesive elements (6-C) with null thicknesses were disposed along the screw shafts of seven bicortical screws, while 548 6-CEs were disposed in the bone shaft to allow damage initiation and propagation in the vicinity of the screw holes. Contact pairs (CP) were also considered in all contacts (i.e., bone plate shaft, plate screw, bone cylinder shaft) to prevent interpenetration in the course of the loading process. 

The load was applied in both numerical models (POS and FPB) under displacement control considering small increments (0.5% of the applied displacement) to induce stable damage growth. Moreover, the boundary conditions were implemented according to the referred experiments, and a nonlinear geometrical analysis was adopted for both simulations.

### Cohesive Zone Modeling

It is obvious that both experimental tests (POS and FPB) induce mixed-mode (I + II) loading in cortical bone tissue. To account for those conditions, cohesive zone modeling considering a trapezoidal bilinear law [16] (Figure 9) is an appropriate tool to mimic damage initiation and crack propagation in bone tissue [16,17].

Before damage initiation, the equivalent traction can be written as
(2)$$\sigma_{m}=k\delta_{m}$$
with k being the interface stiffness and $\delta_{m}$ the equivalent relative displacement,
(3)$$\delta_{m}=\sqrt{\delta_{m,\;I}^{2}+\delta_{m,\;\mathrm{II}}^{2}+\delta_{m,\;\mathrm{III}}^{2}}$$
where $\delta_{m,\;j}$ represent the mode j components (j=I, II, III) of the equivalent mixed-mode I+II+III displacement ($\delta_{m}$). After damage onset, the constitutive Equation (2) alters to account for damage development,
(4)$$\sigma_{m}=\left(1-d_{m}\right)k\delta_{m}$$
where $d_{m}$ is the damage parameter. The determination of the equivalent relative displacements at the critical points ($\delta_{1m}$, $\delta_{2m}$, $\delta_{3m}$, and $\delta_{\mathrm{um}}$ in Figure 9) and equivalent traction at point 3 ($\sigma_{3m}$ in Figure 9) is crucial to establish the mixed-mode softening law via definition of $d_{m}$ in the three softening branches.

Since the compression stresses do not promote damage, the following quadratic stress criterion was used to simulate damage onset [17],
(5)$${\left(\frac{\sigma_{I}}{\sigma_{u,\;I}}\right)}^{2}+{\left(\frac{\sigma_{\mathrm{II}}}{\sigma_{u,\;\mathrm{II}}}\right)}^{2}+{\left(\frac{\sigma_{\mathrm{III}}}{\sigma_{u,\;\mathrm{III}}}\right)}^{2}=1\;\;\;\;\;\;\;\;\mathrm{if}\;\;\sigma_{I}>0$$
$${\left(\frac{\sigma_{\mathrm{II}}}{\sigma_{u,\;\mathrm{II}}}\right)}^{2}+{\left(\frac{\sigma_{\mathrm{III}}}{\sigma_{u,\;\mathrm{III}}}\right)}^{2}=1\;\;\;\;\;\;\;\;\mathrm{if}\;\;\sigma_{I}\leq 0\;\;\;\;\;\;\;\;\;\;\;\;\;\;\;\;\;\;\;\;\;\;\;$$

Combining Equation (2) with the first Equation (5), yields,
(6)$${\left(\frac{\delta_{im,\;I}}{\delta_{1,\;I}}\right)}^{2}+{\left(\frac{\delta_{im,\;\mathrm{II}}}{\delta_{1,\;\mathrm{II}}}\right)}^{2}+{\left(\frac{\delta_{im,\;\mathrm{III}}}{\delta_{1,\;\mathrm{III}}}\right)}^{2}=1\;\;\;\;\mathrm{with}\;\;i=1,\;2,\;3$$

Subscript *i* identifies the inflection points of the cohesive law (Figure 9), and $\delta_{im,\;j\;}$ (j= I,  II,  III) stand for the components of the mixed-mode displacement, $\delta_{m}$. For damage initiation, subscript i=1 and $\delta_{1m,\;j}$ (j=I, II, III) are the displacement components responsible for damage onset. Equation (6) is also employed for the remaining inflection points (2, 3) in the softening region. For these three points, the equivalent relative displacements given by Equation (3) can be rewritten as,
(7)$$\delta_{im}=\frac{\delta_{i,I}\delta_{i,\mathrm{II}}\delta_{i,\mathrm{III}}\sqrt{1+\beta^{2}+\gamma^{2}}}{\sqrt{\delta_{i,\;\mathrm{II}}^{2}\delta_{i,\;\mathrm{III}}^{2}+\beta^{2}\delta_{i,\;I}^{2}\delta_{i,\;\mathrm{III}}^{2}+\gamma^{2}\delta_{i,\;I}^{2}\delta_{i,\;\mathrm{II}}^{2}}}\;\;\;\;\mathrm{with}\;\;i\;=\;1,\;2,\;3$$
being,
(8)$$\beta =\frac{\left|\delta_{\mathrm{II}}\right|}{\delta_{I}}\;\mathrm{and}\;\gamma =\frac{\left|\delta_{\mathrm{III}}\right|}{\delta_{I}}$$

The equivalent traction at the inflection point 3 (Figure 9) is determined using the quadratic stress criterion (Equation (5)), which leads to
(9)$$\sigma_{3m}=\frac{\sigma_{3,I}\;\sigma_{3,\mathrm{II}}\;\sigma_{3,\mathrm{III}}\sqrt{1+\beta^{2}+\gamma^{2}}}{\sqrt{\sigma_{3,\;\mathrm{II}}^{2}\sigma_{3,\;\mathrm{III}}^{2}+\beta^{2}\sigma_{3,\;I}^{2}\sigma_{3,\;\mathrm{III}}^{2}+\gamma^{2}\sigma_{3,\;I}^{2}\sigma_{3,\;\mathrm{II}}^{2}}}$$

The determination of the equivalent ultimate relative displacement ($\delta_{\mathrm{um}}$) is performed by means of the linear energetic fracture criterion
(10)$$\frac{G_{I}}{G_{\mathrm{Ic}}}+\frac{G_{\mathrm{II}}}{G_{\mathrm{IIc}}}+\frac{G_{\mathrm{III}}}{G_{\mathrm{IIIc}}}=1$$
in which $G_{j}$ and $G_{jc}$(with  j=I, II, III) represent, respectively, the strain energy components and the respective critical values that are material properties. This relation can be used to define the ultimate equivalent relative displacement at failure ($\delta_{\mathrm{um}}$). With this aim, the $G_{j}$ components can be obtained from the area of the mode-*j* trapezoid (Figure 9),
(11)$$G_{j}=\frac{\sigma_{3m,\;j}\;\left(\delta_{\mathrm{um},j}-\delta_{2m,j}\right)+k\delta_{1m,j}\;\left(\delta_{2m,j}-\delta_{1m,j}+\delta_{3m,j}\right)}{2}$$
and substituted in Equation (10). Accounting for Equation (3) results,
(12)$$\delta_{\mathrm{um}}=\frac{2G_{\mathrm{Ic}}G_{\mathrm{IIc}}G_{\mathrm{IIIc}}\left(1+\beta^{2}+\gamma^{2}\right)}{\sigma_{3}\left(G_{\mathrm{IIc}}G_{\mathrm{IIIc}}+\beta^{2}G_{\mathrm{Ic}}G_{\mathrm{IIIc}}+\gamma^{2}G_{\mathrm{Ic}}G_{\mathrm{IIc}}\right)}-\frac{k\;\delta_{1m}\left(\delta_{2m}+\delta_{3m}-\delta_{1m}\right)}{\sigma_{3}}+\delta_{2m}$$

The evolution of the damage parameter in the three softening branches of the cohesive law can now be defined as
(13)$$d_{m}=\frac{\left(\delta_{m}-\delta_{1m}\right)}{\delta_{m}}\;\;\;\;\;\mathrm{for}\;\;\;\;\;\delta_{1m}\leq \delta_{m}\leq \delta_{2m}$$
$$d_{m}=1-\frac{1}{k\delta_{m}}\left[\frac{\left(\sigma_{3m}-k\delta_{1m}\right)\left(\delta_{m}-\delta_{2m}\right)}{\delta_{3m}-\delta_{2m}}+k\delta_{1m}\right]\;\;\;\;\;\mathrm{for}\;\;\;\;\;\delta_{2m}\leq \delta_{m}\leq \delta_{3m}$$
$$d_{m}=1-\frac{\sigma_{3m}}{k\delta_{m}}\left(\frac{\delta_{\mathrm{um}}-\delta_{m}}{\delta_{\mathrm{um}}-\delta_{3m}}\right)\;\;\;\;\;\mathrm{for}\;\;\;\;\;\delta_{3m}\leq \delta_{m}\leq \delta_{\mathrm{um}}$$

The damage parameter is subsequently introduced in Equation (4) in order to establish the relation between tractions and relative displacements according to the softening law previously defined.

## 5. Results and Discussion

### 5.1. Experimental Tests

#### 5.1.1. Three-Point Bending Test

The longitudinal elastic modulus ($E_{L}$) of goat cortical bone was obtained from three-point bending tests, taking into account the specimen dimensions (Section 3.1) and the value of bending stiffness (R), according to Equation (1). The obtained result (Table 1) is the mean value, which is very close to the one determined by Li et al. [18]. This value is approximately 83% smaller than the one obtained by Martin et al. [19] for cortical human bone tissue in the femoral region (diaphysis; Haversian bone) in tension. The remaining elastic properties (Table 1) were measured by Pereira et al. [20] and Patil et al. [21].

#### 5.1.2. Pull-Out Strength Tests

Figure 10 shows the load–displacement curves (*P*-*δ* curves) obtained in pull-out tests for four samples (regions) along the femoral shaft (Figure 8a). These curves put into evidence the differences that exist in the mechanical behavior of the bone along the longitudinal axis. It is clear that samples that were harvested from the femoral extremities reveal higher values of initial stiffness and pull-out strength. This behavior is justified by the fact that bone thickness in those regions tends to increase, which is followed by the rise of the trabecular bone along the femoral shaft (see Figure 8a).

#### 5.1.3. Four-Point Bending Tests

Figure 11 shows the set of *P*-*δ* curves obtained in four four-point bending tests of immobilized oblique bone fractures with osteosynthesis metal plates. The scatter observed in these results was justified by natural anatomical differences (i.e., size and configurations) of the bone shafts. Despite that, the initial stiffness was consistent among the tested specimens. The first peak (drop) in the applied load was attributed to the propagation of cracks, which were visible in the course of the loading process in the vicinity of the screws used to fasten the metal plate. 

Following the FPB tests, parts (screws, metal plate, and bone segments) were cautiously disassembled to identify bone-damaged areas with accuracy. Inspection of damaged regions in the bone revealed that cracks invariably initiated near the fracture line (middle diaphysis simple oblique fracture) and propagated in the direction of the first hole (Figure 12). This observation is not surprising since it is known that holes and screw fastening regions induce stress concentrations, leading to damage initiation and propagation in bone cortical tissue. The inspection of damaged regions in the bone (white arrows in Figure 12a,b) was followed by marking bone cracks with ink, which was suitable for further crack-tracking procedures performed numerically. Moreover, screw loosening was visible in most holes, which puts into evidence that the dissipation of energy in the bone–screw interface is not negligible in this fixation procedure.

### 5.2. Numerical Analysis

#### 5.2.1. Pull-Out Tests

Numerical characterization of pull-out screw tests (POS) was performed in four different bone femoral sections (Figure 8a), using the elastic properties presented in Table 1. The determination of cohesive parameters presented in Table 2 was performed by an iterative inverse procedure lying on matching the numerical load–displacement curve with the experimental one for each case [23]. The ensuing results (Figure 13) demonstrate that the experimental curves were well reproduced, which proves that CZM is an adequate way to simulate the pull-out load of bicortical screws.

#### 5.2.2. Four-Point Bending Tests

Numerical analyses of the FPB test were conducted to mimic the experimental results. To this aim, elastic properties of goat cortical bone tissue and used metal parts were considered (Table 1), as well as the set of cohesive parameters obtained in the pull-out tests for each region (Table 2) and the ones corresponding to the bone tissue (Table 3).

Figure 14a,b show the normal stress fields of this test in the longitudinal direction of the bone ($\sigma_{xx}$), in which the normal stress gradient is coherent with the load and boundary conditions imposed in the numerical model. As expected, the stress values are higher in the vicinity of the bone screws and fractured bone regions. The crack propagation occurring during experimental loading in both parts of the bone is also well mimicked through the CZM (red arrows in Figure 14b).

The numerical–experimental agreement for the FPB tests is then reported through *P*-*δ* (load–displacement) curves (Figure 11). The numerical curve replicates the global behavior observed experimentally, reproducing the initial stiffness and the first loading peaks satisfactorily. This leads to the conclusion that the developed model provides a reliable estimation of the damage extent in the vicinity of the used screws and the fractured region of the bone. 

Further analysis of the stress state was performed in the regions sited in the vicinity of each screw on both sides of the bone shaft (Figure 15a), i.e., the ones in contact with the metal plate and on the opposite side. Figure 15b presents the obtained von Mises stresses calculated for each set of elements (S1–S6 and the lag screw: L) forming the cortical bone shaft for the load increment corresponding to the ultimate load (Figure 11). The oblique fracture section was also considered in this study. It can be observed that the maximum von Mises stress component (i.e., 32.84 MPa) occurs at the S1 screw, revealing that this is the critical region for damage development in this fixing procedure.

The profiles of normal and shear stress components obtained in the bone regions sited in the vicinity of screws (S1 to S6 and L) at each side of the bone shaft (i.e., in contact with the metal plate and on the opposite side) are plotted in Figure 16 and Figure 17. Peak values of normal (Figure 16a,b) and shear stresses (Figure 17c,d) were registered for screw S1, with 32.84 MPa (for $\sigma_{zz}$) and 10 MPa (for $\tau_{xz}$), respectively. However, it should be noted that these results are below the threshold that was established in the cohesive zone modeling for both stress values (i.e., normal and shear in Table 2) for the bolted connection S1 (namely, $\sigma_{1,I}=50.0\;\;\mathrm{MPa}$ and $\sigma_{1,\mathrm{III}}=\sigma_{1,\mathrm{II}}=60.0\;\;\mathrm{MPa}$). Moreover, the region affected by the lag screw (L) represents a critical point of this reinforcement system, with $\sigma_{xx}$ and $\tau_{xy}$ stresses attaining −11.85 and 5.13 MPa, respectively. As observed for the remaining bolted regions, the attained stresses in the vicinity of the lag screw are below the cohesive zone strength defined for each bolted connection in Table 2 (the minimum values were 29.3 and 30.0 MPa for $\sigma_{1,I}$, $\sigma_{1,\mathrm{II}}$, and $\sigma_{1,\mathrm{III}}$). In any case, an analysis of Figure 16 and Figure 17 renders it possible to understand how the stress components in the bolted joints of cortical bone vary along the reinforced bone shaft when submitted to bending. It can be observed that S1 and L screws are the most critical ones. A special remark ought to be made for the shear stress component $\tau_{xy}$ in the vicinity of the lag screw since a sliding damage mechanism can occur in this region. However, it must be concluded that the mechanical performance of the bolted joints is quite satisfying in this reinforcement system.

The von Mises stresses were also analyzed in the oblique fracture section (Figure 18). It can be observed that the stress field at the inferior part of the oblique section is almost null, due to the separation of bone sections in the course of the bending loading. Since bone healing requires the existence of compressive stresses along the fracture section, it can be concluded that this is a critical region regarding the restorative process. This stress distribution is governed by the applied load, which can be estimated with the developed numerical model. 

Bone fracture repairing techniques can benefit a lot from the employment of numerical models that allow simulating damage initiation and propagation in bone tissue. This strategy originates the accurate evaluation of bone stress (including the fracture surfaces), rendering it possible to analyze the effect of bone repair therapies. Once applied to human bone tissue, the developed procedures may contribute to help surgeons to support decisions regarding bone repair using standard DCP and, posteriorly, locking plates.

## 6. Conclusions

Femoral shafts of healthy Serrana breed goats (animal models) were harvested to evaluate the mechanical behavior of a current solution used by surgeons to immobilize oblique bone fractures by dynamic compression plates (DCP). Finite element modeling using stress and strain failure criteria was employed. The strain criterion was adopted to simulate the interaction of a typical bone configuration with stainless-steel DCP with cortical screws submitted to four-point bending tests. The estimated maximum values for stress were made considering the adopted plate configuration in an idealized bone trauma. Some fixation solutions were found rather bulky, requiring careful adaptation to the bone surface and space. As screw loosening was visible in the course of the bending process, the evaluation of pull-out strength (POS) of cortical screws proved to be essential to evaluate the mechanical performance with accuracy. A numerical approach based on mixed-mode I+II+III cohesive zone modeling was adopted using realistic configurations of the employed bone samples for different regions of the bone. The obtained cohesive parameters were then used in realistic numerical models to replicate the experimental results obtained in immobilization systems employed in simple oblique bone fractures repaired with DCP. The developed model was able to mimic both the screw loosening phenomenon and damage (cracks) in the vicinity of the screws. A detailed stress analysis in the regions of the screws and oblique fracture surface allowed identifying the critical regions prone to damage development and restorative difficulties. It can be concluded that the developed procedure can be used as a valuable numerical tool to help surgeons to support decisions regarding bone repair using standard DCP and, posteriorly, locking plates.

## Figures and Tables

**Figure 1 biology-11-00940-f001:**
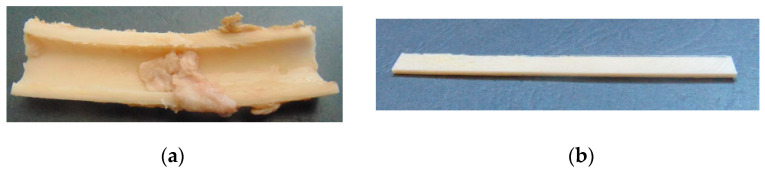
(**a**) Cortical bone after the execution of the longitudinal cut; (**b**) TPB specimen.

**Figure 2 biology-11-00940-f002:**
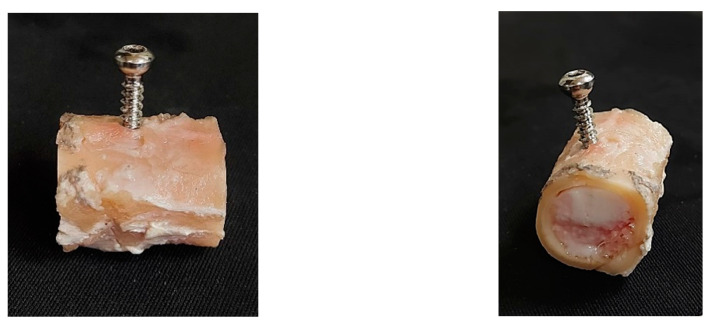
Bone samples for POS tests.

**Figure 3 biology-11-00940-f003:**
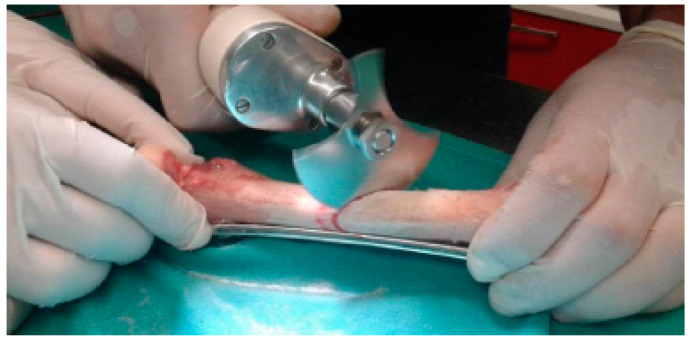
Fabrication of the oblique fracture.

**Figure 4 biology-11-00940-f004:**
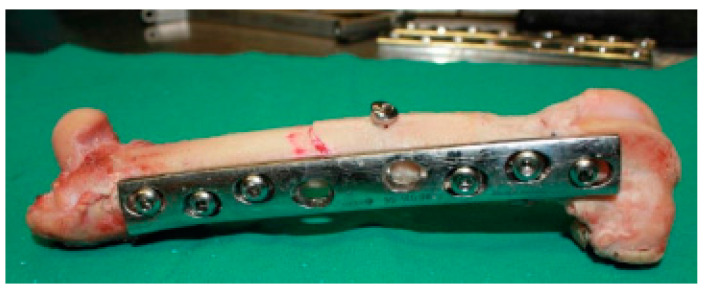
Fixation of the DCP plate to the femur shaft.

**Figure 5 biology-11-00940-f005:**
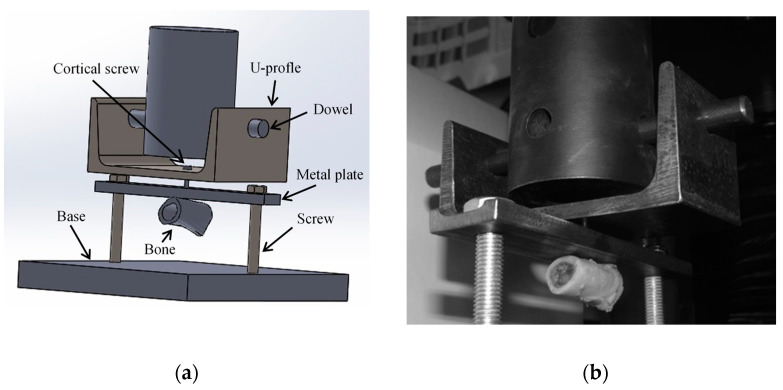
Schematic representation (**a**) and pull-out strength test (**b**) in the goat cortical bone.

**Figure 6 biology-11-00940-f006:**
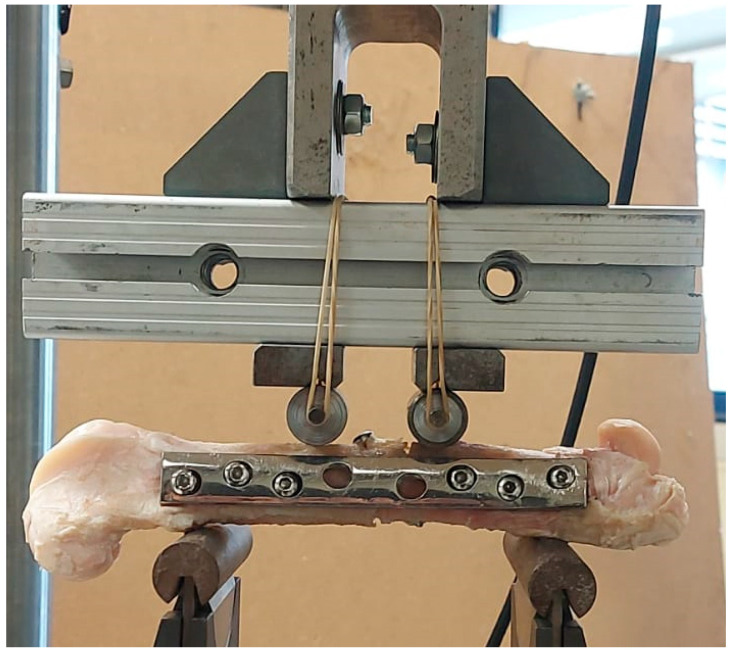
Experimental setup of FPB tests.

**Figure 7 biology-11-00940-f007:**
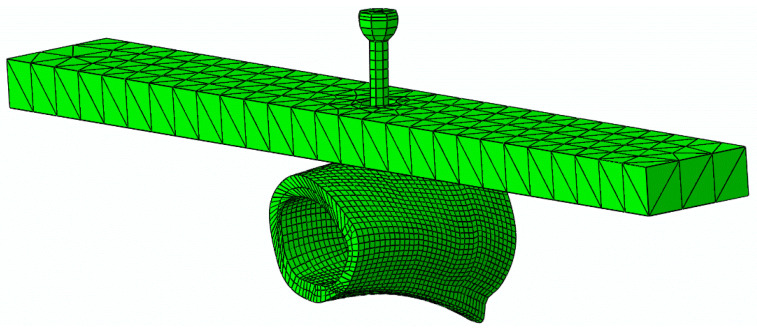
Finite element mesh of the pull-out screw (POS) test.

**Figure 8 biology-11-00940-f008:**
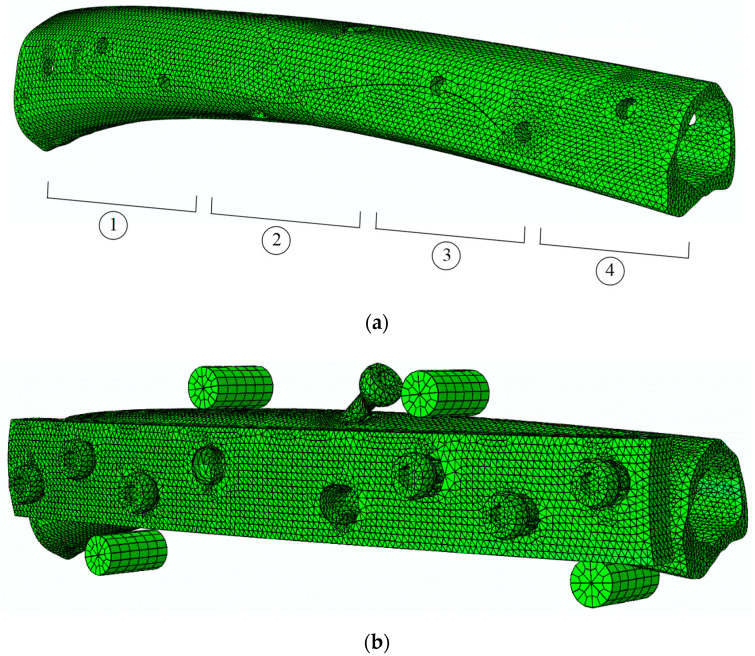
Finite element mesh (**a**) of the bone shaft (approximately 30 mm in length each) showing four (1–4) regions to perform pull-out strength tests, and (**b**) the four-point bending (FPB) tests.

**Figure 9 biology-11-00940-f009:**
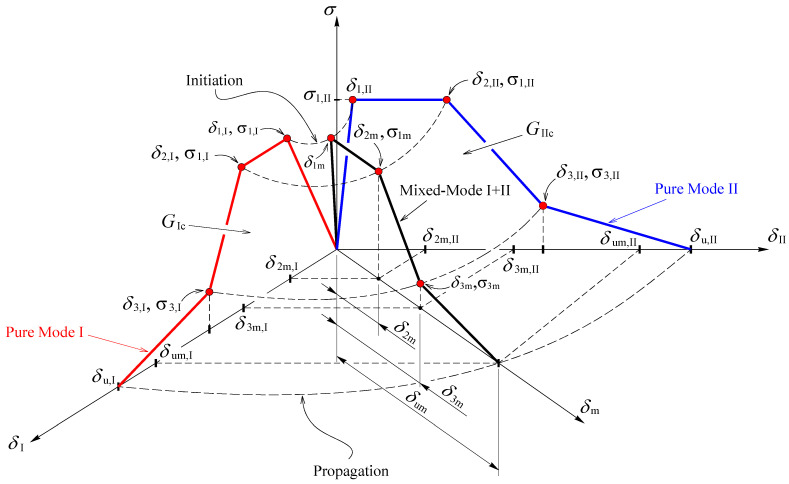
Mixed-mode I+II cohesive law.

**Figure 10 biology-11-00940-f010:**
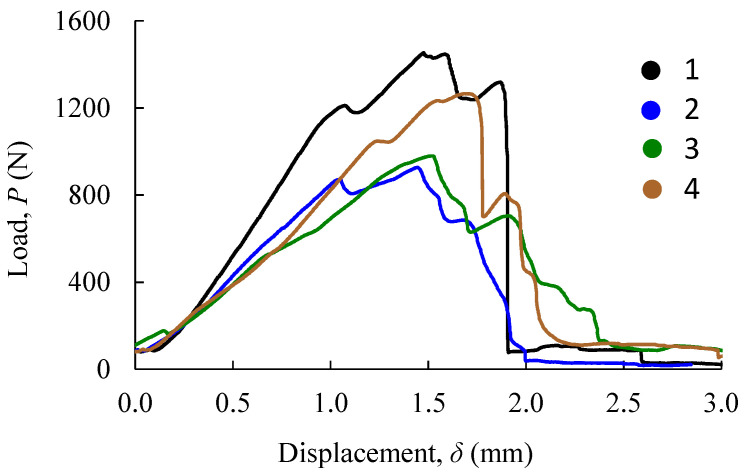
*P*-*δ* curves obtained in pull-out tests in four samples (regions).

**Figure 11 biology-11-00940-f011:**
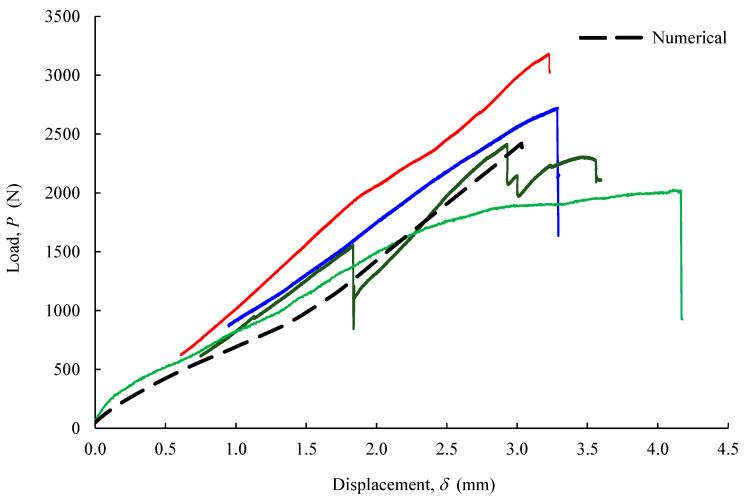
FPB tests (load–displacement) experimental curves.

**Figure 12 biology-11-00940-f012:**
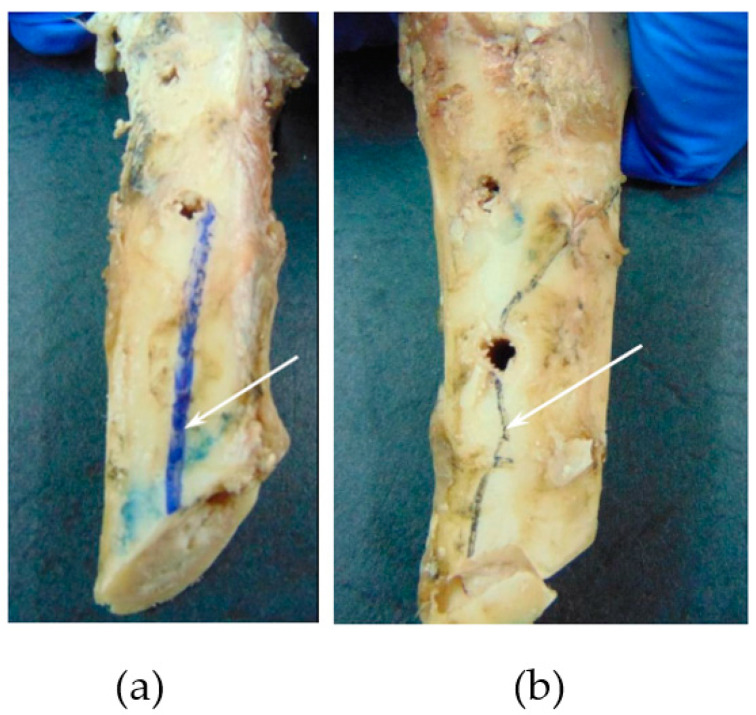
Fracture propagation, including (**a**) proximal epiphysis and (**b**) distal epiphysis.

**Figure 13 biology-11-00940-f013:**
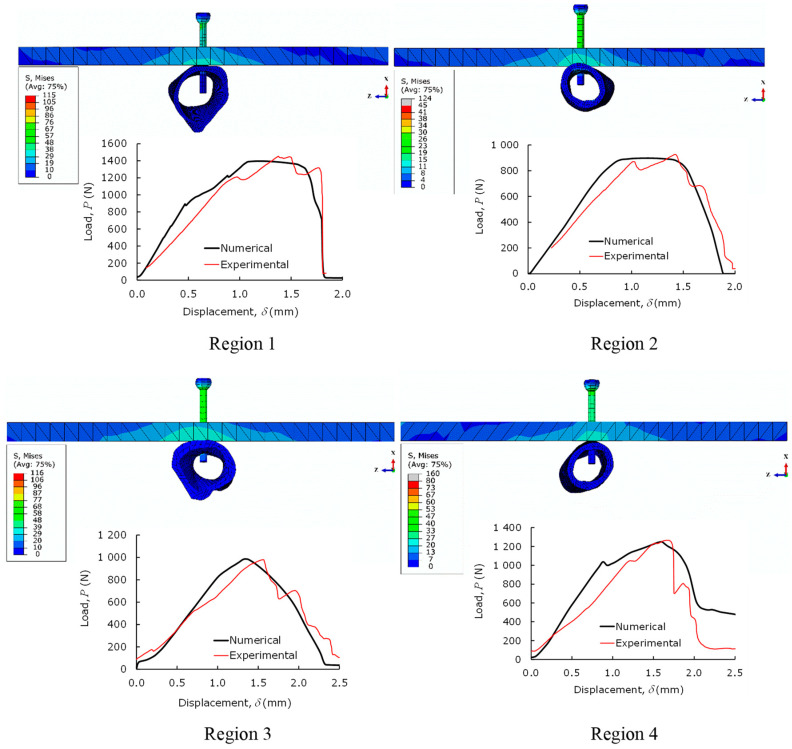
Numerical agreement of pull-out tests for each region shown in Figure 8a.

**Figure 14 biology-11-00940-f014:**
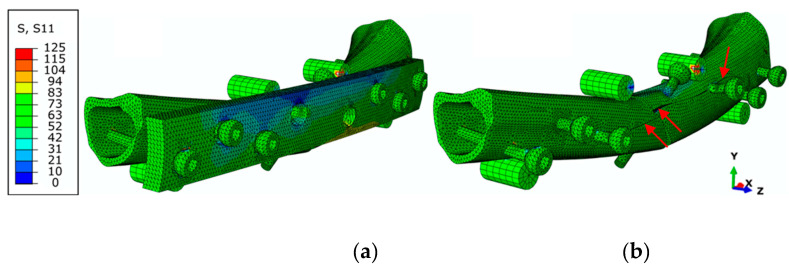
Normal stress field ($\sigma_{xx}$) (in MPa) of the FPB finite element model showing (**a**) the DCP plate and (**b**) when hiding the DCP plate (red arrows show bone cracks in bone).

**Figure 15 biology-11-00940-f015:**
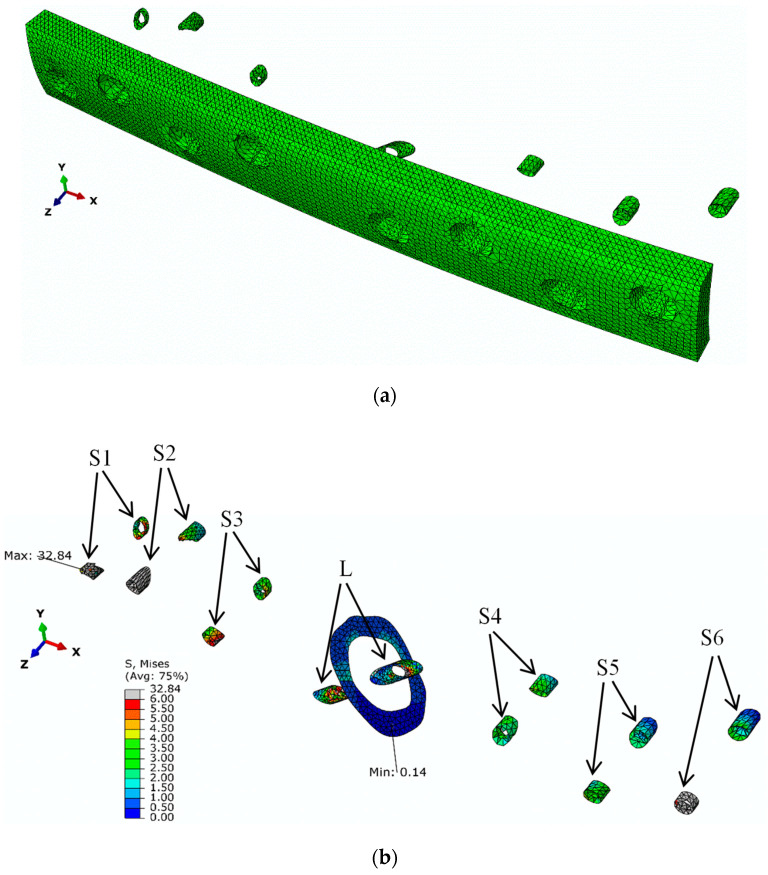
(**a**) Identification of the elements selected for stress analysis; (**b**) von Mises stresses at the seven screws (S1–S6 and the lag screw: L) and at the oblique fracture surface.

**Figure 16 biology-11-00940-f016:**
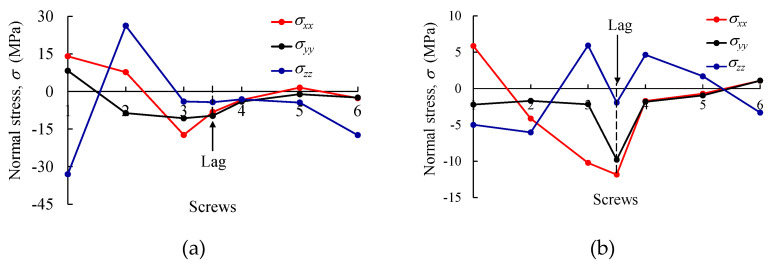
Normal stress in bone regions S1S6 (and Lag) sited (**a**) in contact with the metal plate and (**b**) on the opposite side of the bone shaft.

**Figure 17 biology-11-00940-f017:**
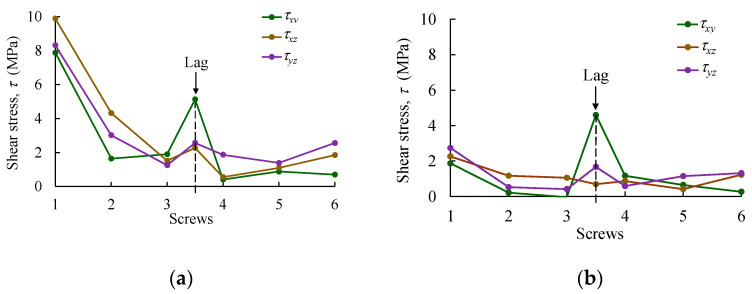
Shear stress in bone regions S1–S6 (and Lag) sited (**a**) in contact with the metal plate and (**b**) on the opposite side of the bone shaft.

**Figure 18 biology-11-00940-f018:**
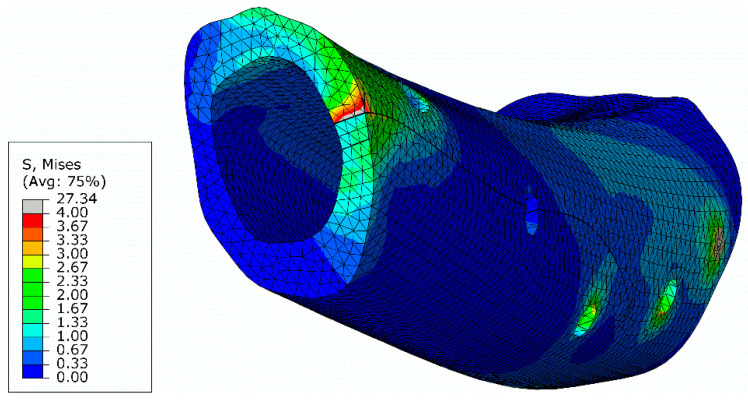
Stresses distribution (MPa) at the lag screw and at the oblique fracture surface.

**Table 1 biology-11-00940-t001:** Elastic properties used in numerical simulations.

Materials	Elastic Properties	
**Goat Cortical Bone Tissue**	$E_{L}$	14.8 GPa ^(1)^
	$E_{R}=E_{T}$	8.9 GPa ^(2)^
	$\upsilon_{\mathrm{LT}}=\upsilon_{\mathrm{LR}}$	0.17 ^(3)^
	$\upsilon_{\mathrm{RT}}$	0.18 ^(3)^
	$G_{\mathrm{RT}}$	3.60 GPa ^(3)^
	$G_{\mathrm{RL}}=G_{\mathrm{LT}}$	3.57 GPa ^(3)^
**316L Stainless Steel (316L SS)**	E	193 GPa ^(4)^
	υ	0.28 ^(4)^

^(1)^ Obtained in this study; ^(2)^ in [20]; ^(3)^ in [21]; ^(4)^ in [22].

**Table 2 biology-11-00940-t002:** Parameters of the cohesive mixed-mode (I+II) law obtained in pull-out tests.

Sec.	$G_{\mathrm{Ic}}$ (N/mm)	$G_{\mathrm{IIIc}}\;=\;G_{\mathrm{IIc}}$ (N/mm)	$\sigma_{1,I}$ (MPa)	$\sigma_{1,\mathrm{III}}\;=\;\sigma_{1,\mathrm{II}}$ (MPa)	$\delta_{2,I}$ (mm)	$\delta_{2,\mathrm{III}}\;=\;\delta_{2,\mathrm{II}}$ (mm)	$\sigma_{3,I}$ (MPa)	$\sigma_{3,\mathrm{III}}\;=\;\sigma_{3,\mathrm{II}}$ (MPa)	$\delta_{3,I}$ (mm)	$\delta_{3,\mathrm{III}}\;=\;\delta_{3,\mathrm{II}}$ (mm)
1	2.00	70.0	50.0	60	0.035	1.0	30.10	24.0	0.039	1.200
2	1.63	45.0	29.3	35	0.050	0.7	18.83	12.6	0.054	1.450
3	1.63	35.0	70.0	33	0.010	0.4	43.73	11.27	0.020	1.270
4	1.63	41.3	110.0	30	0.001	0.1	74.32	8.97	0.010	1.890

**Table 3 biology-11-00940-t003:** Parameters of the cohesive mixed-mode (I+II) law for the goat cortical bone obtained by the inverse procedure detailed in [23].

$G_{\mathrm{Ic}}$ (N/mm)	$G_{\mathrm{IIIc}}\;=\;G_{\mathrm{IIc}}$ (N/mm)	$\sigma_{1,I}$ (MPa)	$\sigma_{1,\mathrm{III}}\;=\;\sigma_{1,\mathrm{II}}$ (MPa)	$\delta_{2,I}$ (mm)	$\delta_{2,\mathrm{III}}\;=\;\delta_{2,\mathrm{II}}$ (mm)	$\sigma_{3,I}$ (MPa)	$\sigma_{3,\mathrm{III}}\;=\;\sigma_{3,\mathrm{II}}$ (MPa)	$\delta_{3,I}$ (mm)	$\delta_{3,\mathrm{III}}\;=\;\delta_{3,\mathrm{II}}$ (mm)
1.5	2.0	20.0	20.0	0.01	0.01	10.0	10.0	0.075	0.1

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
