# Peer review of "Osteosynthesis Metal Plate System for Bone Fixation Using Bicortical Screws: Numerical–Experimental Characterization"

_biology, 2022, doi:10.3390/biology11060940_

Round 1
Reviewer 1 Report
Dear Authors: I want to congratulate with You for the study, that is clearly well conducted. Nevertheless, results have been well known for decades, and currently DCPs are no longer used in clinical practice, given the presence of newer plates (LC-DCPs, LCPs, VA-LCPs, just to name plating system of the same manufacturer) . The added value of this study is the finite elements study. I will recommend the paper to be accepted after revision (English editing is required), and re-consideration about the real actual value of the study. best regards,
Author Response
The Authors would like to thank the Reviewer for his/her appreciation. As suggested, the English was reviewed and improved.
In an initial phase, we chose to carry out this study using standard DCP. However, in the near future, we intend to carry out other ex vivo studies using the most current locking plates, and even other types of plates from other manufacturers with mechanical principles of fracture stabilization different from DCP and LCP.
Reviewer 2 Report
This is a very valuable study combining experiments, modelling and numerical analysis. The paper is well written; the rationale behind the activity is clear and well explained, and the results are clearly presented.
There are just some very minor issues to be addressed to authors:
- Check the name of the last author: “Nuno D and rado” ?
- How did you build the virtual models of bone after mechanical testing?
- Line 247: why bold text?
- Insert a reference for Equation 5, if it is taken from previous works.
- Adding basic statistical analysis could be considered by authors.
Author Response
Reviewer comments: This is a very valuable study combining experiments, modelling and
numerical analysis. The paper is well written; the rationale behind the activity is clear and well
explained, and the results are clearly presented
Our reaction: The Authors thank the Reviewer for his/her positive comments on our work.
Query 1: Check the name of the last author: “Nuno D and rado”?
Answer: Many thanks for your attentive reading. This error has been corrected.
Query 2: How did you build the virtual models of bone after mechanical testing?
Answer: The explanation of this issue has been improved, including more details and discussion.
The procedure inherent to virtual models building has been made performing successive parallel
(very) fine cuts along the longitudinal axis of the bone shaft, subsequent to mechanical tests. Solid
models were first constructed from digital images of bone cross sections following the original
orientation of bone. Three-dimensional models (CAD) were elaborated using home-made
numerical tools developed in MatLab®, from a sequence of contours of the referred
cross-sections.
This explanation and discussion have been made in Section 4 of the revised manuscript.
Query 3: Line 247: why bold text?
Answer: Many thanks for the attentive reading. The mistake was corrected.
Query 4: Insert a reference for Equation 5, if it is taken from previous works.
Answer: A reference was duly inserted as requested by the Reviewer.
Query 5: Adding basic statistical analysis could be considered by authors.
Answer: The objective of this work was to present and validate a methodology applicable to bone
fracture repair using osteosynthesis plates. In this context, the number of performed tests was not
sufficient to undergo a systematic material behavior, which would justify the suggested statistical
analysis. In fact, the main goal was only to detail suitable numerical and experimental procedures
that can be used in systematic studies involving a significant number of experimental results. This
has not been done in this work, but is envisaged for near future.
Reviewer 3 Report
I would like to thank to editors and the authors for the opportunity to have read such a quality original research paper.
This original research article aims to test the behavior of the bone fracture stabilized with dynamic compression plate in a neutralization function, associated with a lag screw on an experimental femoral diaphysis model. The non-linear behavior was tested using four-point bending test and screw loosening was tested. A numerical model and a finite model were also analyzed. The developed procedure is valuable and can help the clinician decide on which hardware is needed.
The paper is very well documented, and the research is well written.
We recommend the publication of the paper in its current form.
Author Response
The Authors would like to thank the Reviewer for his/her appreciation. We really appreciate the positive and encouraging comments.
Reviewer 4 Report
The authors present a study that focuses on mechanical behaviour of a standard immobilization system that is currently used to treat oblique fractures of femural diaphyses. The sudy is well documented and presents a numerical as well as an experimental characterization.
Author Response

(The authors gave the same response as above.)

Round 2
Reviewer 1 Report
Dear Authors: after review, and answers to my questions, I recommend the paper to be published in the present form. best regards, and congrats.